# Regulation of Nuclear Mechanics and the Impact on DNA Damage

**DOI:** 10.3390/ijms22063178

**Published:** 2021-03-20

**Authors:** Ália dos Santos, Christopher P. Toseland

**Affiliations:** 1Department of Oncology and Metabolism, University of Sheffield, Sheffield S10 2RX, UK; 2Insigneo Institute for in Silico Medicine, University of Sheffield, Sheffield S10 2RX, UK

**Keywords:** mechanics, DNA, chromatin, nucleus, lamin, cytoskeleton, DNA damage

## Abstract

In eukaryotic cells, the nucleus houses the genomic material of the cell. The physical properties of the nucleus and its ability to sense external mechanical cues are tightly linked to the regulation of cellular events, such as gene expression. Nuclear mechanics and morphology are altered in many diseases such as cancer and premature ageing syndromes. Therefore, it is important to understand how different components contribute to nuclear processes, organisation and mechanics, and how they are misregulated in disease. Although, over the years, studies have focused on the nuclear lamina—a mesh of intermediate filament proteins residing between the chromatin and the nuclear membrane—there is growing evidence that chromatin structure and factors that regulate chromatin organisation are essential contributors to the physical properties of the nucleus. Here, we review the main structural components that contribute to the mechanical properties of the nucleus, with particular emphasis on chromatin structure. We also provide an example of how nuclear stiffness can both impact and be affected by cellular processes such as DNA damage and repair.

## 1. Introduction

The nucleus houses the genetic information necessary for the activity and survival of the cell, but as we outline in this review, the nucleus is more than just a compartment to house DNA. Although the nucleus is the largest and stiffest cellular organelle, it is also a highly dynamic organelle that can sense the external environment and rapidly adapt [1,2,3]. The nuclear envelope comprises a double membrane—the outer nuclear membrane and the inner nuclear membrane—associated with various distinct transmembrane proteins, such as nuclear pore complexes and LEM (Lap2, emerin, and Man1)-domain proteins [4]. This is followed by an assembly of lamin filaments at the nuclear interior that provides structural stability to the organelle and tether chromatin to the nuclear envelope. From the outside, the nucleus is linked to the cytoskeleton through the Linker of Nucleoskleton and Cytoskeleton (LINC) complex, which also binds to the nuclear lamina [5]. This nuclear connectivity allows external signals to modulate nuclear functions, such as transcription [6,7] and DNA replication [8,9]. Moreover, it may allow communication in the opposite direction (Figure 1).

Within the nucleus, the DNA associates to histone cores to form nucleosomes, the building blocks of chromatin. Through epigenetic regulation, chromatin can be packaged into different conformations and higher-order structures, which determine the accessibility [10] of DNA to replication [11,12], transcription [13,14], and repair machinery [15,16]. Higher-order compact chromatin structures, known as heterochromatin, are largely inaccessible and are usually associated with genomic regions of low transcriptional activity at the nuclear periphery [17,18,19]. Meanwhile, more open conformations of chromatin, also known as euchromatin, are easily accessible and represent areas of active gene expression [20,21].

Variations to the biochemical components of the nucleus result in changes to the physical properties of the organelle and its morphology. The nuclear mechanical properties, comprising the viscoelastic behaviour and plasticity, are tightly linked to cellular function and vary between cell stages and types [22]. There are four major contributors to nuclear shape and the mechanical properties: the magnitude of cytoskeletal forces exerted on the organelle, the composition and thickness of the nuclear lamina, the level of chromatin compaction within the nucleus, and the activity of proteins that modulate DNA structure (Figure 2).

Altered nuclear morphology and mechanics are usually accompanied by changes in gene expression and cell function. Changes in the shape and size of the nucleus have been reported for different diseases, and in some cases, this can also be used to help diagnosis. For example, abnormally shaped nuclei can be found in cardiomyopathies, progeria and in cancer. In particular, nuclei of cervical cancer cells present herniations or blebbing, and this constitutes part of the Pap smear test diagnosis [23]; in breast cancer, nuclear pleomorphisms (altered nuclear morphology) are used for tumour grading and correlates with patient outcome [24]. It is therefore essential to understand how these changes in nuclear morphology arise, how they reflect altered mechanical properties of the nucleus and how this affects overall cellular function, mechanosensing and force transduction.

Here, we pay special attention to the newly emerging data on the importance of chromatin dynamics and the regulation of its spatial organisation. We also discuss some new technological approaches in mechanobiology and in the study of chromatin architecture. Finally, we will discuss how nuclear mechanics can influence cellular processes such as DNA damage.

## 2. Contributing Factors to Nuclear Mechanics

### 2.1. Cytoskeletal Forces in Nuclear Mechanics

Mechanotransduction refers to the process by which cells respond to external mechanical cues through the activation of biochemical pathways, changes in structure, and activation or repression of specific genes. This is a key mechanism for sensing and adapting to the extracellular microenvironment. The nucleus is the largest and the most mechanically prominent organelle in the cell, and so it would be expected to play a dominant role in cellular mechanics. It is, therefore, not surprising that it has been receiving increasing attention in the field of cell mechanics over the last decade.

Extracellular forces propagate into the nucleus through the LINC complex, located on the nuclear envelope. The LINC complex physically connects the nucleus to the cytoskeleton and, therefore, to cellular adhesions that can sense the mechanical microenvironment. Disruption of the LINC complex results in defective force transduction from the cytoplasm to the nucleus and is shown to disrupt the expression of mechanosensitive genes [25,26]. This occurs, at least in part, because cytoskeletal forces can directly affect the localisation and nuclear import of mechanosensitive transcription regulators. One functional example is the Yes-associated protein (YAP) and its transcriptional coactivator with PDZ-binding motif (TAZ). The mechanical regulation of YAP/TAZ requires cytoskeletal integrity and a functional LINC complex [27]. As a result, translocation of YAP/TAZ to the nucleus and consequent activation of YAP-dependent genes varies according to extracellular matrix rigidity [28,29], cytoskeletal formation [30], LINC complex integrity [31], and nuclear stiffness [27,32].

Cytoskeletal forces also contribute to the regulation of nuclear movement, shape, orientation and morphology. In extreme cases, the accumulation of aberrant actin fibres around the nucleus can induce actin-dependent nuclear deformation through increased nuclear tension and lead to blebbing, herniation and even rupture of the nuclear envelope [33]. Equally, chemical or genetic perturbation of the cytoskeleton results in deficient force transmission to the nucleus.

The tissue microenvironment is an important factor in cytoskeletal formation and structure, and hence it is a key determinant of the mechanical forces transmitted to the nucleus. This is particularly important when studying cancer biology and therapeutic approaches to disease. We are beginning to have insight into the mechanical consequences of different microenvironments and how they affect nuclear processes such as DNA repair, transcription and chromatin regulation.

However, studying the effects of cytoskeletal forces on nuclear processes comes with specific challenges. Many studies use drugs to disrupt cytoskeletal forces, and it is unclear how these may directly affect nuclear events, such as transcription or chromatin organisation. Latrunculin B, for example, disrupts cytoskeletal formation through inhibition of actin polymerisation, but it also disrupts nuclear actin and myosin functions, directly affecting transcription levels and RNA Polymerase II (RNAPII) spatial organisation [34]. Methods such as micropatterning [35,36] or the use of gels as substrates for cells [37,38,39] are becoming increasingly common to change the stresses exerted upon the cytoskeleton. Although these do not allow complete disruption of the cytoskeleton, they can mimic different tissue microenvironments and permit the fine-tuning of cytoskeletal forces. The data are both easier to interpret and, in some cases, can be a good alternative to chemical disruption of the cytoskeleton.

Accumulating evidence shows that actomyosin contractility around the nucleus has not only important repercussions to the deformability of the organelle, and therefore to cell migration and nuclear integrity, but it can also directly affect lamina structure and chromatin organisation. For example, using micropatterned substrates, Makhija and colleagues observed that cells lacking long stress fibres had reduced levels of lamin A/C expression, resulting in softer, more deformable nuclei. Interestingly, these cells also displayed increased chromatin and telomere dynamics, suggesting a direct relationship between geometric cell constraints and genome organisation [40]. Similarly, in rod photoreceptor cells, actomyosin deformation of the nucleus results in chromocenter clustering (condensed centromere heterochromatin regions) and inverted chromatin architecture, with euchromatin regions being redistributed to the nuclear periphery [41].

Although the molecular mechanisms that link cytoskeletal forces to chromatin regulation are still largely unexplored, spatial redistribution and misregulation of nuclear transport of transcription factors and chromatin regulators, such as transcription coactivator MKL1 [42] or histone deacetylase HDAC3 [43], have been observed and could account, at least in part, for chromatin architectural changes.

A challenge often encountered in the study of nuclear mechanics—especially when performing measurements on adhered cells—is the masking effect that the stiff cytoskeletal fibres around the nucleus have on measurements. Indeed, small changes in nuclear mechanics can be imperceptible if measured under a fully-formed cytoskeleton. Because of this, many studies resort to the use of isolated nuclei, removing the cytoskeletal impact entirely. This has proven useful when comparing nuclei across different cell types [44] to study how the expression of nuclear envelope components affects nuclear stiffness [45], or how the nucleus itself—and independently of cytosolic intervention—adapts and responds to external forces [46]. The clear trade-off is the loss of physiological environment and understanding how nuclei in live-cells respond to matrix stiffness, drug treatments, radiation or other challenges. To overcome these limitations, dos Santos et al. have recently performed mechanical measurements, using atomic force microscopy (AFM), in live cells, but at initially adhered stages. In this case, cells were allowed sufficient time to attach to their surface but, with their cytoskeleton not yet fully formed, the nucleus then occupies most of the cell volume and is the most important contributor to cell mechanics. This has allowed the detection of mechanical changes that would have otherwise not been observed [47].

Overall, there is a complex interplay of interactions between the nucleus and cytoskeletal components that contribute to the mechanics of the organelle, and much work is needed to understand how changes in cytoskeletal forces directly affect nuclear organisation and nuclear processes.

### 2.2. The Nuclear Lamina 

The nuclear lamina is located between the INM and the chromatin. This is a dense, complex meshwork of proteins with a thickness up to 100 nm. It provides major structural support to the nucleus as well as support for a variety of nuclear functions, such as transcription, replication, DNA repair, and genome organisation. Lamina proteins fall into two separate classes, A-type and B type—lamin A and C, which are splice-isoforms of the *LMNA* gene—belong to the former and lamin B1 and B2, encoded by *LMNB1* and *LMNB2*, respectively, belong to the latter [48]. Lamins belong to a class of proteins called intermediate filaments, which contain rod domains that are critical to the formation of the meshwork [49].

Post-translational modifications of lamin proteins allow the regulation of this peripheral meshwork. One important post-translational modification is the farnesylation of both lamin A and B at their C-terminal domains, which is thought to be important for the localisation and retention of these proteins at the nuclear envelope. Whilst lamin B is permanently farnesylated, lamin A is further processed by proteolytic cleavage, which includes removal of this group [48,50,51].

Whilst B-type lamins are expressed throughout development and in all nucleated cells, the levels of type-A lamins are reduced or not present at early embryonic stages. Expression of lamin A onset is highly varied in different tissues during development and, in some cases, such as for stem cells or cells of the hemopoietic system, lamin A is never expressed [52].

The nuclear lamina is crucial for maintaining nuclear envelope integrity. Depletion or mutation of lamin components leads to severe nuclear instability, morphological defects in the nucleus and gives rise to disease, as in the case of laminopathies. For instance, in mice, loss of either lamin B1 or B2 leads to neuronal defects and perinatal death [53,54]. Similarly, in humans, mutations in lamin A are associated with premature ageing and muscular malfunction, as observed in Hutchinson–Gilford progeria, muscular dystrophy, and cardiomyopathies [55].

The morphological defects observed in the nucleus of cells with lamina defects are also indicative of the important role of these proteins in nuclear mechanics. This is not surprising, as expression levels of lamin proteins, but in particular of lamin A/C, scale with nuclear stiffness. Depletion of lamin A/C makes the nucleus softer and more deformable, whilst expression of a shorter and permanently farnesylated mutant version, progerin, confers higher stiffness to the nucleus. In both cases, the expression of mechano-responsive genes is severely disrupted [22,56]. Importantly, in these cells, nuclear processes such as chromatin structure regulation, DNA replication, DNA repair and gene expression are also invariably misregulated.

However, the nuclear envelope is not mechanically isolated. Instead, it is physically connected to the cytoskeleton on one side, through the LINC complex, and to the chromatin, through lamina-associated domains (LADs) on the other. This means that mechanical changes to one of these components have large structural implications for the others. For example, Swift et al. described how matrix and cytoskeletal stiffness could directly influence lamin A expression and turnover [57]. In stiffer matrices, phosphorylation of lamin A, which promotes disassembly, is reduced, and this increases total amounts of lamin A at the nuclear envelope, which in turn increases nuclear stiffness [57,58]. Similarly, progerin-expressing cells, such as those from Hutchinson–Gilford progeria patients, display increased Filamentous-actin (F-actin) polymerisation and cytoskeletal stiffness, as well as reduced levels of heterochromatin [59]. In these cells, destabilisation of microtubules to reduce cytoskeletal tension may be a promising therapeutic approach. It has been shown to restore nuclear morphology and alleviate premature ageing in progeria in mice [60]. Conversely, a separate study also described how inhibition of histone demethylation, which directly leads to increased heterochromatin levels, also rescues morphological defects in progerin-expressing cells [61].

Chromatin function is highly dependent on its conformation. This includes correct tethering to LADs at the nuclear periphery. Loss of lamin A function, for example, leads to higher chromatin dynamics and more diffuse genomic loci, representative of higher levels of decondensed chromatin. Complete lamin loss results in detachment of LADs and disruption of global 3D chromatin-chromatin interactions [62]. Similarly, lamin B1 also has an important role in maintaining chromatin structure and distribution, especially at the nuclear periphery [63]. As a result, lamin regulation and chromatin structure are tightly linked as key components of nuclear mechanics. Future insights into how changes to nuclear envelope components affect the mechanical properties of the nucleus will be especially important in the study of ageing and cancer, where lamina mutations are often found.

### 2.3. Chromatin Is a Key Component of Nuclear Mechanics 

Chromatin organisation is highly regulated through epigenetic histone modifications that determine local and global levels of DNA compaction. The degree of chromatin condensation and the nuclear content of hetero versus euchromatin affects not only nuclear size and morphology but also determines DNA accessibility to transcription machinery and all forms of DNA processing. In conventional nuclei, the highly compacted heterochromatin is spatially segregated from active, decondensed euchromatin, with the former usually occupying regions in the nuclear periphery and the latter in the nuclear core. Developments in genome-wide chromosome conformation capture (Hi-C) have shown that in addition to LAD formation, chromatin can also associate with itself to form sub-compartments called topologically associated domains (TADs) [64,65].

We now know that chromatin is also a major contributor to nuclear stiffness and morphology. Historically, research has focused on the more clear-cut structural contributions of lamins, in particular, lamin A. Early experiments using micropipetting showed that the lamina dominates the mechanics of swollen nuclei, whilst chromatin is the main contributor to the mechanics of shrunken nuclei. Although this suggested two different types of mechanical contribution in the nucleus, the chromatin is often regarded as a minor viscous component that flows upon applied force [22,66,67]. This view of chromatin as a secondary mechanical component, less important than the lamina, has started to change, and some recent studies have now shed light on how chromatin architecture can affect nuclear stiffness. A recent report by Strickfaden et al. highlights this by showing that self-associated condensed chromatin behaves as a solid or elastic gel instead of a liquid. This indicates that the intrinsic mechanical properties of chromatin have an impact on overall nuclear mechanics and its response to external force stimuli [68].

In line with this, experiments by Chalut et al., using an optical stretching technique, show how nuclear deformability is directly related to the degree of chromatin condensation [69]. Furthermore, experiments using MNase, for chromatin digestion also show that its structure determines nuclear morphology and governs nuclear responses to short-extension strains (<30%) [70]. We now know that changes in nuclear mechanics and shape, including the occurrence of nuclear blebbing, can occur independently of lamin perturbation due to changes in the levels in euchromatin and heterochromatin. Overexpression of nucleosome binding protein HMGN5 leads to chromatin decondensation, increased nuclear area, morphological aberrations to the nuclear envelope and nuclear softening. In mice, this leads to premature death as a result of cardiac defects [71]. Similarly, treatment with deacetylase inhibitor trichostatin A, which alters histone modifications and leads to chromatin decondensation, also results in nuclear softening and blebbing [47,61,72].

Trying to resolve individual mechanical contributions from the lamina and chromatin is a major challenge for the field of nuclear mechanics. Their physical linkage means that force measurements provide a composite value for the contribution of both components. New methodologies in force measurements could prove particularly useful. For instance, a combined AFM-side-view light-sheet fluorescence microscope developed by Nelsen et al. allowed the 3D (*x/z* cross-section) visualisation of live-cells whilst performing force measurements with high spatio-temporal resolution [73]. In fact, Hobson et al. recently used this approach to propose a two-regime nucleus, where lamina and chromatin respond differently to volume deformation and nuclear-area stretching [73]. Using 3D fluorescent imaging of the nucleus combined with mechanical measurements could be a powerful tool not only to probe the mechanical behaviour of different nuclear components but also to look at force transduction from the cytoskeleton, or how nuclear processes, such as DNA repair, affect cell mechanics.

Whilst mechanical differences between the lamina and chromatin are readily intuitive, differences within the chromatin itself at the nuclear interior are less so. Chromatin architecture is not homogeneous; its organisation is highly regulated, with different domains occupying different regions. This means that the nuclear interior is mechanically non-uniform. Using AFM microrheology on isolated nuclei, Lherbette et al. were able to detect mechanical variations within the nucleus, representative of a largely inhomogeneous chromatin interior. The authors observed two different mechanical regions within the nucleus, independently of the nuclear lamina—a more viscous periphery and a stiffer and predominantly elastic nuclear core [44]. A new methodology that allows a more in-depth study of chromatin environments could be important to understand how changes to chromatin organisation influence the mechanical behaviour of the nucleus.

As chromatin becomes more prominent in the field of nuclear mechanics, it becomes obvious that processes and proteins that regulate DNA structure and conformation have a large impact on the mechanical properties of the nucleus. As will be discussed later, DNA damage, occurring either through ionising radiation or genotoxic agents, such as chemotherapy drugs, causes large genomic alterations, including changes to chromatin architecture and transcription levels [74]. This can result in global relaxation of the chromatin and mechanical softening of the nuclear envelope [47]. Our knowledge of how these processes affect nuclear mechanics is still limited but may be of crucial importance.

### 2.4. Chromatin Conformation and Crosslinking in Nuclear Mechanics 

In the nucleus, chromatin associates with a large variety of DNA-binding proteins that regulate its structure and spatial organisation. Examples of this are the previously mentioned lamins that allow crosslinking of the chromatin to the nuclear periphery. This is achieved through interactions with lamina-associated proteins and is important in the regulation of global chromatin structure. Chromatin conformations and mobility are irregular throughout the nucleus and largely determined by its level of crosslinking.

Using liquid chromatin Hi-C, Belaghzal and colleagues recently showed that chromatin loci dynamics and association are largely determined by chromatin-associated factors, such as cohesins and lamins. The authors found that chromatin behaves as a crosslinked polymer gel that, even upon digestion (within 10–25 kb), maintains its structural and mechanical connections. Furthermore, chromatin digestion at this scale did not affect nuclear stiffness, measured by micropipetting. Interestingly, after extensive chromatin digestion (<6 kb), loss of chromosome compartmentalisation was achieved, together with loss of chromatin-associated cohesins, which resulted in a 75% decrease in nuclear stiffness [75]. This indicates that compartmental segregation of chromosomes and nuclear mechanics are highly dependent on the crosslinking capabilities of proteins that modulate DNA structure.

Cohesins are highly conserved protein complexes that can loop chromatin through their ring domain to create bundles that restrict chromatin movement [76,77]. Together with the chromatin insulator CTCF, cohesin function is the driver behind TAD formation and hence crucial for chromatin 3D organisation [78,79]. CTCF/cohesin anchoring of chromatin has been shown to be important at different genomic length-scales, allowing the formation of long-range Topologically Associating Domain (TAD) compartments (megabase-sized), as well as intermediate (100 kb–1 Mb) and small-range (<100 kb) sub-compartments [80,81]. Heterochromatin protein 1 (HP1) is another protein that is known for its role in chromatin organisation. HP1 binds to H3K9me3-rich areas, which represent constitutive heterochromatin regions [82], and is capable of bridging nucleosomes [83]. This crosslinking effect of HP1 is thought to stabilise compacted chromatin states and to be essential in heterochromatin phase separation [84,85] through the formation of membrane-less condensates. A recent report by Strom et al. also showed that HP1 chromatin crosslinking capabilities are important for nuclear shape maintenance, and its degradation leads to decreased chromatin stiffness and nuclear rigidity [86].

Chromatin crosslinking also occurs at active chromatin regions. For example, the transcriptional coactivator BRD4 can create condensates at active super-enhancer regions [87]. Similarly, during transcription, RNAPII forms large molecular clusters with transcription factors. These networks have not only been associated with phase separation events but also with the formation of transient chromatin bridges [88]. Interestingly, whilst active chromatin regions are usually associated with open and dynamic conformations, recent reports challenge this view by showing that the bridging effect of RNAPII and associated transcription machinery can increase chromatin constraints, reduce chromatin mobility and possibly affect local stiffness [89,90].

The arising field of nuclear actin and myosins also provides an interesting new perspective in chromatin mechanics. Actin and myosins, in the cytoplasm, are usually associated with structural and mechanical roles. Whilst actin is one of the most abundant cytoplasmic proteins; its nuclear levels are comparatively low and tightly regulated—actin is actively transported in and out of the nucleus by importin-9 and exportin-6, respectively. Due to the low concentrations of actin and the highly dynamic and transient nature of nuclear actin filaments, visualisation of these structures is particularly challenging, and much debate still surrounds them.

There is, however, growing evidence that nuclear F-actin assembly plays important roles in transcription [91], mitosis [92], DNA replication [93], DNA repair [94] and chromatin regulation [95]. Nuclear F-actin filament formation has been observed following DNA repair response (DDR) activation, caused by telomere uncapping, treatment with genotoxic drugs or UV radiation. F-actin is recruited to sites of damage, and the formation of polymeric actin structures appear to be necessary for repair factors to cluster at DNA lesions [96,97,98,99]. These observations followed early studies showing the actin-dependent intranuclear movement of genomic loci.

To visualise nuclear F-actin polymers, Baarlink and co-workers developed a phalloidin-based assay and used dSTORM super-resolution imaging. The authors observed that these structures are necessary for reshaping nuclei following mitosis. Depletion of nuclear actin filaments, either by overexpression of exportin-6 or a polymerisation-deficient mutant of nuclear actin (NLS-actin^R62D^), prevented the usual chromatin decondensation and nuclear expansion that occurs following cell division. In addition, this also led to impaired transcription and, in the case of mouse embryos, impaired development [92].

Budding yeast experiments support the idea of a role for nuclear F-actin in chromatin organisation. In this case, inhibition of actin polymerisation was shown to reduce telomere and chromosome dynamics, and, interestingly, it also reduced the efficiency of DNA repair by homologous recombination (HR) [100].

It seems very likely that mechanical changes to the nucleus would occur upon the formation of nuclear actin filaments. In the cytoplasm, the extent of actin fibre formation scales with cellular stiffness. However, it is still unclear how nuclear actin filament formation affects nuclear mechanics and nuclear mechanosensing. Nuclear actin may induce direct changes to local mechanics or alternatively through altered chromatin compaction and dynamics.

In the cytoplasm, the role of actin filaments is tightly linked to the function of myosins. These molecular motors use ATP hydrolysis to generate force and movement, and thus, are essential in several cellular processes such as cell migration [101], intracellular transport [102,103] and membrane regulation [104]. Similar to F-actin, these proteins are also present in the nucleus. However, there are still many unanswered questions regarding their nuclear functions.

One example is Myosin VI (MVI). MVI associates with RNAPII and is important in transcription regulation [105,106,107]. However, the specific molecular roles of MVI in the nucleus and how they might be connected to nuclear actin filament formation have only recently started to come to light. A report by Zorca et al. showed that MVI is necessary for chromatin reorganisation at the early stages of transcription. In this study, the authors found that inhibition of MVI or actin prevented allelic pairing and gene repositioning after transcription stimulation [108]. This suggests MVI has the ability to reorganise and move chromatin across long nuclear distances and also to crosslink chromatin to allow gene proximity during transcription. More recent work by Große-Berkenbusch et al. allowed single-molecule tracking of MVI along actin filaments in the nucleus of HeLa cells [109]. This movement was observed across several micrometres, resembling the cytoplasmic motility of this molecular motor. The authors also reported ATPase-depend movement of MVI on chromatin in vitro and described how this could have an important function in chromatin organisation by regulating long-range chromatin movement. In agreement with this, using STORM imaging, Hari-Gupta et al. have shown that MVI regulates the spatial organisation of RNA Polymerase II (RNAPII) during transcription initiation. Both disruption of nuclear actin polymerisation and MVI force-sensing—through the introduction of an MVI spring mutant that changes its response to force—abrogate RNAPII distribution and have severe consequences for gene expression using RNA-Seq [34]. The role of MVI in regulating RNAPII clustering also suggests a role for this molecular motor in RNAPII-dependent chromatin bridging and droplet formation [89,90], thus leading to local changes to chromatin mechanics.

Another myosin, nuclear myosin I (NMI), also appears to be involved in a plethora of nuclear roles, including chromatin movement and DNA repair. NMI is a component of the chromatin remodelling complex B-WICH, which allows chromosomal rearrangement for RNAPI-related transcription [110,111]. Recently, KO experiments of NMI in mouse embryonic fibroblasts led to increased levels of heterochromatin and lower levels of active chromatin markers such as H3K9ac [112]. Interestingly, another study showed that with cisplatin damage, NMI is recruited to the chromatin and facilitates chromosome territory relocation in an ATM and γH2AX-dependent manner [113].

Work connecting the activity of NMI, actin and the nuclear lamina also reinforce the idea that these proteins might have more extensive structural and mechanical roles in the nucleus than previously thought [114,115,116]. As these proteins are well-known for their role in force transduction in the cytoplasm, they may have a similar role in the nucleus [117,118]. Future studies are necessary to shed light on the response of nuclear F-actin and nuclear myosins to external mechanical cues.

## 3. The Relationship between DNA Damage and Nuclear Mechanics

Having described the contributing factors to nuclear mechanics, we will now exemplify how this can impact cellular processes using DNA damage. DNA damage events continuously occur throughout the life cycle of a cell. Following DNA damage, the cell activates the DNA damage response (DDR) to ensure appropriate repair of lesions and their survival. Failure within these mechanisms leads to apoptosis or drives genomic instability leading to diseases such as cancer. Both DNA damage and the DDR lead to large nuclear reorganisation and activation of different biochemical pathways that may result in changes to the mechanical properties of the nucleus.

DNA damage can arise endogenously as a result of replication errors [119,120], topoisomerase activity [121] or reactive oxygen species (ROS) [122]. Conversely, external insults to the cell, such as ionising radiation or chemotherapy drugs, also cause significant DNA damage and genomic instability. Types of damage range from base-mismatches, adducts and crosslinks introduced into the DNA helix, single-strand DNA breaks or, the more deleterious and toxic, double-strand breaks. In order to allow efficient repair and prevent the propagation of mutations to daughter cells, once a lesion is detected, DDR is activated, cell cycle progression is halted, and transcriptional activity becomes markedly reduced. This is accompanied by changes to the 3D organisation of chromatin.

Two key DDR components in mammalian cells are the Ataxia Telangiectasia-mutated (ATM) and the ATM Rad3-related protein (ATR) kinases. These early responders are recruited to sites of damage to initiate DNA damage checkpoints and are responsible for the phosphorylation of additional DNA repair factors that recruit and regulate repair machinery [123]. Histone modifications also have a major role in DDR. An example is histone H2AX, which following double-strand break detection is phosphorylated (γH2AX) close to the lesion site. This creates repair centres, known as repair foci, which act as signalling hubs for the rapid recruitment of repair factors.

DNA damage and repair are not independent of the chromatin architecture. Two very early studies highlighted this relationship by showing that more accessible, open structures of chromatin are more susceptible to nuclease digestion [124,125]. We now know that local chromatin dynamics change following DNA damage and how this may be important for DDR [126]. For example, Hauer et al. describe how induction of a single double-strand break in yeast increases damaged chromatin flexibility and motility as a result of chromatin decondensation and histone eviction [127]. This was dependent on the activity of ATP-dependent chromatin remodelling complex INO80. Similarly, work in both yeast and mammalian cells show the recruitment of different remodelling complexes, such Swi/Snf family members, and changes to overall chromatin structure during DDR [128,129,130,131,132,133]. Chromatin relaxation is now considered to be an important part of DDR that improves repair efficiency.

The chromatin context and nuclear positioning of the DNA lesion are also crucial determinants for DDR pathway selection [134,135,136]. Lesions occurring in active transcription regions of euchromatin, associated with histone mark H3K36me3, are preferentially repaired by the error-free HR [137,138]. For these regions, the major determinant in pathway selection appears to be chromatin accessibility and not transcriptional activity [139,140,141]. Paradoxically, DNA breaks in heterochromatin, which packages inactive transcription sites, can also preferentially undergo HR [142,143,144,145]. However, for this to occur, DNA lesions need to be relocated to more permissive environments. Chiolo et al. described that, in *Drosophila,* following a double-strand break, the heterochromatin domain where the break occurs expands, and the lesion is relocated to the periphery or to the exterior of this domain [146]. This movement of DNA lesions to euchromatin regions allows access to repair machinery and enables HR [146,147]. Similar dynamics of double-strand breaks have also been observed in yeast [148,149,150]. However, this is not always the case, and there is a large variability between the repair of lesions in different heterochromatin environments. For DNA breaks at LADs, HR is restricted to favour error-prone non-homologous end-joining, and, in this case, lesions do not migrate to euchromatin regions [139]. Similarly, differences between repair in centromeric and pericentric heterochromatin have also been reported, suggesting other factors might also influence repair strategies [151]. Although chromatin relaxation and reorganisation following DNA damage has been well documented, very little is known about how this impacts nuclear structure and mechanics. To shed light on this question, dos Santos et al. recently used AFM to probe the mechanical properties of the nucleus in cisplatin-treated cells. This work showed that DNA damage leads to significant softening of the nucleus, and this was the direct result of global chromatin decondensation [47]. Reduction in nuclear stiffness correlated with the levels of damage and was ATM-dependent, suggesting that DDR activation is necessary. In line with previous work, the chromatin decondensation observed led to more rapid molecular diffusion in the nucleus, which could lead to higher accessibility of repair factors and improve repair efficiency (Figure 3). Interestingly, a recent report also revealed a role for the ATR kinase in nuclear mechanics. Kidiyoor et al. showed that ATR ensures coupling of the nuclear envelope to the cytoskeleton, and its depletion leads to changes in chromatin organisation, nuclear softening and reduction in nuclear circularity [152].

The consequences of DDR-induced nuclear softening on mechanotransduction are still unclear. However, it is likely that mechanical changes to the organelle during repair will have repercussions for cell migration, force transduction to the cytoplasm and transcription regulation.

The mechanical properties of the cell and the nucleus prior to the damage, are also known to influence the extent of DNA damage that the cell suffers, as well as the outcome and efficiency of repair. Increased cytoskeletal stiffness can lead to higher levels of DNA damage in the cell [153], and this could be particularly important during confined migration. When cells migrate, especially through spaces smaller than their cross-sectional area, substantial compression forces are exerted on the nucleus. The nuclear deformation caused by these forces can in itself cause DNA damage, but it can also lead to rupture of the nuclear envelope, which, by exposing the nuclear contents to cytoplasmic factors, can cause severe genomic instability [154,155]. Furthermore, during migration, the deformation of the nuclear envelope, as well as its rupture, can cause local loss of repair factors and other mobile proteins and have severe implications to the efficiency of DDR pathways [156].

Both the cytoskeletal forces exerted on the nucleus, and nuclear stiffness, are important in determining nuclear deformability and the propensity of the nuclear envelope for rupture. As a result, the mechanical properties of the nucleus are tightly linked to the levels of DNA damage caused by migration and the outcome of DDR. Softer nuclei, with reduced levels of lamin A/C, require less pressure for nuclear envelope rupture to occur during migration and, therefore, display increased levels of DNA damage [154,156,157]. Similarly, in experiments using trichostatin A, nuclear softening caused by chromatin decompaction increased the likelihood of nuclear envelope rupture after shear stress application via syringe passes [61].

Although this work indicates that nuclear stiffening can have a protective role for the cell, cells expressing progerin, which increases nuclear stiffness, display higher levels of DNA damage; the mechanisms through which this occurs are not entirely clear, especially as progerin-expressing cells appear to require higher forces for nuclear envelope rupture [158]. However, this reinforces the idea of a link between altered nuclear mechanics and the incidence of DNA damage in the cell. Interestingly, cells treated with cisplatin display different levels of DNA damage, depending on the stiffness of their substrate. Using polyacrylamide gels of different stiffness, dos Santos et al. showed that on softer surfaces, where the nuclear tension is lower due to reduced cytoskeletal forces, cells have lower levels of DNA damage marker γH2AX following cisplatin treatment. This is also true for cells treated with blebbistatin prior to DNA damage induction [47]. The reasons for this are not yet clear, and much work is still needed in order to understand how the mechanical state of the nucleus relates to the propensity of the cells for DNA damage. Furthermore, how the pre-existing mechanical state of the nucleus might influence the DDR pathway choice is also not clear. As we discussed, changes in cytoskeletal forces and lamina composition affect nuclear stiffness and are also likely to affect chromatin organisation. Similarly, dysregulation of chromatin compaction due to, for example, mutations in chromatin remodelling complexes are often observed in cancer, but there are no data on how this may affect overall nuclear mechanics. It is perhaps possible that, as nuclear mechanics is reflective of the chromatin environment, nuclear stiffness might influence DDR pathway choice and, as a result, genomic stability. Understanding this relationship is important, particularly in therapeutics, since the mechanical properties of the cells and of the nucleus are often altered in disease, and a mechanically compromised nucleus could mean a different outcome in terms of drug efficacy.

## 4. Conclusions

The physical properties of the nucleus, including its shape and viscoelastic behaviour, are a result of the combined contribution of cytoskeletal forces, nuclear envelope composition, chromatin structure and its regulation by DNA-binding factors.

Cytoskeletal tension exerted on the nucleus directly affects nuclear stability and can alter nuclear envelope components, such as lamin A expression and turnover. Originally, the nuclear lamina was seen as the major contributor to nuclear rigidity and structure. Its composition and thickness directly correlate with nuclear stiffness and affect nuclear processes, including transcription and the spatial organisation of chromatin. Recently, chromatin has emerged as a key contributor to nuclear mechanics. We now know that chromatin behaves as a polymer gel that responds to mechanical stimuli and contributes to overall nuclear stiffness. Chromatin architecture is tightly regulated in cells but is often disrupted in diseases, such as cancer. This is usually accompanied by changes in nuclear morphology and genomic instability.

Cellular processes such as the DNA damage response and repair are known to cause changes in chromatin organisation, and this can alter the mechanical properties of the nucleus. Although changes to chromatin structure that occur during DDR lead to more efficient repair, which may drive chemotherapeutic resistance, it is still unclear how the resulting mechanical changes to the nucleus affect cellular mechanosensing, migration, and cell survival.

Overall, understanding how the mechanical properties of the nucleus affect cellular processes will lead to a better understanding of disease and ageing, along with the development of targeted therapeutic strategies.

## Figures and Tables

**Figure 1 ijms-22-03178-f001:**
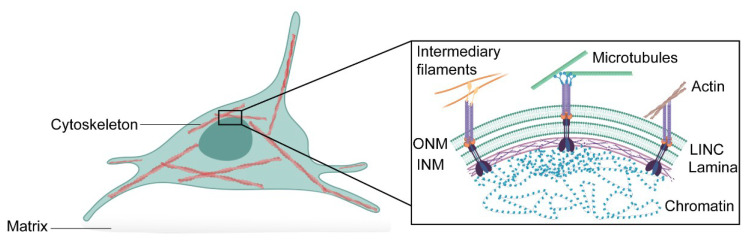
**Schematic representation of the interconnectivity between cytoskeleton, nuclear envelope and chromatin.** The cytoskeleton is physically connected to the nuclear envelope consisting of the outer nuclear membrane (ONM) and the inner nuclear membrane (INM) through the LINC complex. The LINC complex is formed of trimers of SUN-domain proteins that bind different KASH-domain proteins at the nuclear membrane. LINC complexes can indirectly associate with intermediary filaments and microtubules through cyto-linker proteins or motor proteins, respectively, or directly interact with actin filaments. At the nuclear interior, the nuclear lamina tethers chromatin domains—lamina-associated domains—to the nuclear envelope. This allows effective mechanotransduction in the cell.

**Figure 2 ijms-22-03178-f002:**
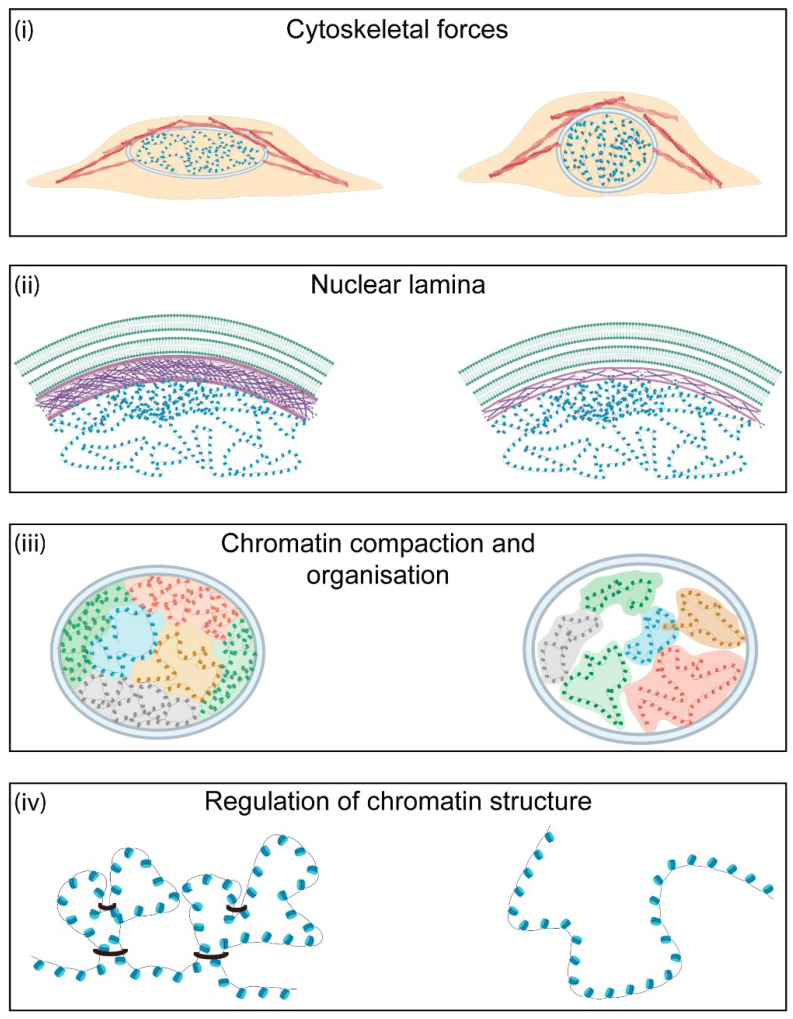
**Major contributors to nuclear morphology and mechanics.** There are four major contributors to nuclear mechanics in the cell. (i) Cytoskeletal forces determine nuclear shape and morphology. Increased actin polymerisation leads to higher nuclear tension. (ii) The nuclear lamina is one of the major contributors to nuclear stiffness. This meshwork of intermediary filaments at the nuclear periphery is important for nuclear stability and chromatin organisation. Higher levels of lamin A/C or a thicker nuclear lamina lead to increased nuclear stiffness. (iii) Chromatin behaves as a crosslinked polymer gel. As a result, changes in chromatin organisation and levels of heterochromatin and euchromatin directly affect nuclear shape and mechanics. (iv) Regulation of chromatin architecture is dependent on the activity of several factors, such as cohesins, that allow crosslinking of chromatin and the formation of higher-order structures. The activity of these proteins can affect local and global chromatin conformation and, hence, is important for nuclear mechanics.

**Figure 3 ijms-22-03178-f003:**
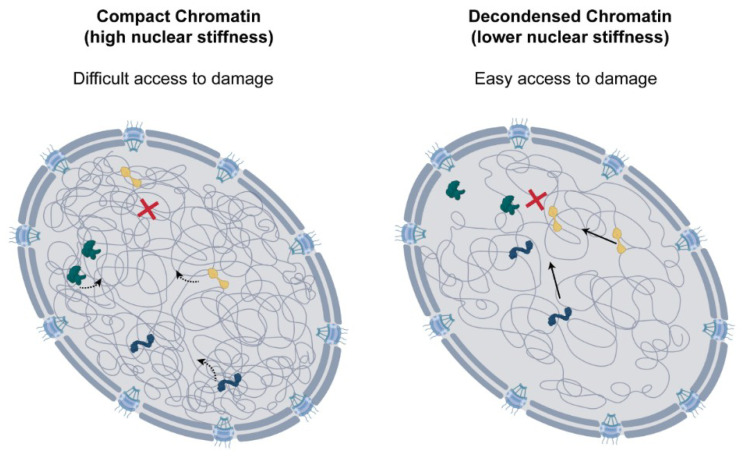
**Role of chromatin decondensation following DNA damage.** Chromatin decondensation leads to higher nuclear diffusion, which could be important for rapid access of repair factors to DNA lesions. This could support the efficient repair of DNA damage.

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
