# Peer review of "Regulation of Nuclear Mechanics and the Impact on DNA Damage"

_ijms, 2021, doi:10.3390/ijms22063178_

Round 1

Reviewer 1 Report

In this review, Santos et al discuss factors that contribute to the mechanical properties of the cell nucleus – (1) cytoskeletal forces, (2) the nuclear lamina and associated NE proteins, with much emphasis on (3) chromatin and (4) proteins regulating chromatin structure and organization.  Their discussion on the physical properties of various aspects of nuclear function display great breadth and depth.  In particular, the authors provide a refreshing perspective of chromatin and its associated proteins and how genome organization and activities impact nuclear mechanics both directly and indirectly by modulating, for example, the levels of nuclear lamins.  In this regard, the authors may have succeeded in stimulating the curiosity in those who only associate the genome with genic activities and who often are not thinking about the mechanical consequences of its organization.  A good point was the recognition of the caveats in the current literature throughout their discussion, as well as recognizing the present technological limitations.  The review ends with a discussion on DDR to exemplify the concepts put forth and were quite thought provoking- tying in changes to general chromatin compaction during damage repair and nuclear pliability, and how this may impact cell migration that has the tendency of inducing nuclear ruptures.  While general chromatin relaxation in the face of genotoxic agents may be true with regards to the genome as a whole, it would be interesting to comment on the degree of (any) relaxation of damage occurring in different nuclear subcompartments undergoing different repair pathways (PMIDs 24658350, 24931610, 25366693, 21353298, 27397684) e.g. DDR at the lamina versus chromocenters and within euchromatin.  A short discussion with some conjecture of how mechanical forces may influence these repair pathways would also be an interesting and provocative addition to this review article.

Author Response

We thank the reviewer for their comments and suggestions. We have now incorporated further discussion regarding DDR with respect to nuclear position and chromatin context.

Reviewer 2 Report

This is an excellent review paper. This nicely summarized the factors and mechanisms that generate the mechanical forces on the nucleus. The writing is clear. So, I have only a few minor comments as described below.

Minor comments:

  1. page 2, the second paragraph: I couldn't understand the meaning of the first sentence, “variations of the physical components of the nucleus result in changes to the material properties of the organelle and its morphology.” In particular, I couldn't imagine what the terms “physical” and “material” mean in the sentence. Do you want to say “variations of the “biochemical” components of the nucleus result in changes to the “physical” or “mechanical” properties of the organelle and its morphology”? If I am wrong, keep it as is or revise this sentence appropriately.
  2. page 4, the fifth paragraph: “Methods such as micropatterning or the -------- increasing common.” This sentence gives a logically abrupt impression. It may be necessary to devise to clarify the relationship with the upper part.
  3. page 8, the second paragraph, lines 5-6: It contains an unnecessary return (indent) after the word “transient.” Remove it.
  4. page 8, the fourth paragraph: Add the word “DNA” before “replication”.
  5. page 9, the third paragraph, line 10: The word “Myosin VI (MVI)” appeared here can be abbreviated to “MVI” because it appears in the first line of the paragraph.
  6. References: I suggest revising this section as follows: 2. “**This ---1941” can be omitted. 13, 16, 22, 73, 91, 95, 106 and 107. The PNAS papers must have the "USA" notation at the end.
  7. The text is on the right side of the page. On the other hand, the figures are on the left. Make sure this layout is correct.

Author Response

We thank the reviewer for their comments and edits. We have made these changes throughout the manuscript.

Reviewer 3 Report

With their manuscript dos Santos and Toseland provide an interesting review on nuclear mechanics. By contrast to many other reviews on this topic they put more emphasis on the role of the chromatin in nuclear mechanics and focus less on the nuclear lamina. I also like that the review mentions a couple of experimental caveats when working on this topic. These are certainly a good reasons to publish this review. I have only a few critical remarks that could easily be addressed in a revised version.

  1. There is a lack of references in the general introduction on the first two pages.

  2. One should also mention here that the LINC complex also binds to the lamin network.

  3. It should be made clearer that the nuclear lamina consists of lamins and nuclear envelope transmembrane proteins (NETs).

  4. One very good example for mechanotransduction in transcription factor function is YAP signaling. This should be mentioned somewhere.

  5. Figure 1: the left part of the inset should refer to intermediate filaments, I think. If so, it should be labeled accordingly and it should be avoided to use the same filament symbols that for actin on the right. Of course it is obvious that the figure is simplified, it should be mentioned in the legend that the LINC complex employs different KASH domain proteins for the association with the different cytoskeletal elements and that KASH interactions with MTs and IFs are indirect.

  6. p. 5, second half: for the understanding of the differential roles of A- and B-type lamins in mechanotransduction it is important to note that only B-type lamins are prenylated permanently. This distinguishes progerin from lamin A.

  7. Reviews are of general interest also for readers outside our field of research. Having this in mind, I feel that there are too many abbreviations. Abbreviations generally hamper readability of texts and here there are no length regulations here which would enforce their extensive use. TSA for example appears only two times. It is also better to write "double strand breaks" instead of DSBs or "nuclear envelope" instead of NE, just to give examples.

Author Response

We thank the reviewer for their comments and suggestions which improve the manuscript. We incorporated these changes throughout the manuscript.